# Extraction and Quantification of Azelaic Acid from Different Wheat Samples (*Triticum durum Desf.*) and Evaluation of Their Antimicrobial and Antioxidant Activities

**DOI:** 10.3390/molecules28052134

**Published:** 2023-02-24

**Authors:** Chiara Spaggiari, Giannamaria Annunziato, Costanza Spadini, Sabrina Lucia Montanaro, Mattia Iannarelli, Clotilde Silvia Cabassi, Gabriele Costantino

**Affiliations:** 1Department of Food and Drug, University of Parma, 43124 Parma, Italy; 2B.Ethical srl SB, Parco Area delle Scienze, 43124 Parma, Italy; 3Department of Veterinary Science, University of Parma, 43126 Parma, Italy

**Keywords:** wheat, azelaic acid, plants extracts, analytical methods, green techniques

## Abstract

Azelaic Acid (AzA) is a 9-carbon atom dicarboxylic acid, with numerous pharmacological uses in dermatology. Its effectiveness in papulopustular rosacea and acne vulgaris, among other dermatological disorders such as keratinization and hyper-pigmentation, is thought to be related to its anti-inflammatory and antimicrobial properties. It is a by-product of *Pityrosporum* fungal mycelia metabolism but also it is found in different cereals such as barley, wheat, and rye. Diverse topical formulations of AzA exist in commerce, and it is mainly produced via chemical synthesis. In this study we describe the extraction of AzA from whole grains and whole-grain flour (*Triticum durum Desf.*) through green methods. Seventeen different extracts were prepared and analyzed for their AzA content by HPLC-MS methods and then screened for their antioxidant activity using spectrophotometric assays (ABTS, DPPH, and Folin–Ciocalteu). Minimum-inhibitory-concentration (MIC) assays against several bacterial and fungal pathogens were performed, to validate their antimicrobial activity. The obtained results indicate that whole grain extracts provide a wider spectrum of activity than the flour matrix; in particular, the Naviglio^®^ extract showed higher AzA content, while the hydroalcoholic ultrasound-assisted extract provided better antimicrobial and antioxidant activity. The data analysis was performed using principal component analysis (PCA), as an unsupervised-pattern-recognition technique, to extract useful analytical and biological information.

## 1. Introduction

Azelaic acid (AzA) is a saturated C_9_ dicarboxylic acid which is produced by a variety of plants in response to biotic- and abiotic-stress conditions [1], thus acting as a natural inducer of plant defence through priming mechanisms. AzA is also produced by several microorganisms, including *Malassezia furfur* [2]. AzA possesses several biological activities, especially when applied topically. Studies have shown its efficacy in the treatment of acne vulgaris, rosacea, and various hyperpigmentation disorders. AzA has been shown to decrease the inflammatory cascade of cytokines, IL-1, IL-6, and TNF-α through the inhibition of nuclear transcription factors NF. The reduction of inflammation, coupled with anti-keratinizing and antibacterial effects, renders AzA an effective topical treatment for mild to moderate acne and other chronic skin diseases such as rosacea, which affects the cheeks, nose, and eyes [2]. The follicular microflora of acne lesions is mainly dominated by *Propionibacterium acnes, Staphylococcus epidermidis,* and *Malassezia furfur* which are inhibited by AzA, but actions against several other bacteria, including *Staphylococcus aureus, Escherichia coli,* and *Pseudomonas aeruginosa* have been also reported. Moreover, a study showed that AzA’s antibacterial activity is associated with a change in bacterial intracellular pH [3]. AzA is also an anti-keratinizing agent; treatment with AzA has been shown to lead to an alteration of epidermal keratinization, particularly affecting the terminal phases of epidermal differentiation, with a corresponding reduction in the size and number of keratohyalin granules. In summary, after treatment with AzA, the infundibular epidermis of acne subjects shows a marked reduction in inflammation and a normalization of filaggrin distribution, as well as comedones containing fewer bacteria and spores. Because of its proprieties, AzA is used for different therapeutic purposes: for the treatment of acne vulgaris and rosacea, hyperpigmentation, and in general as an antibacterial; also, it is formulated in many readily available medical and cosmetical devices, such as serum, gel, and creams. The difference between cosmetic and medical devices is the concentration of the active ingredient in the formulation. The concentration of AzA in topical products is between 10% (cosmetics) and 20% (medical devices/pharmaceuticals) [4]. 

Pharmaceutical and cosmetic formulations, as well as medical devices, use synthetically derived AzA [4]. The strong quest for production methods which are an alternative to organic synthesis and based on the principle of green chemistry prompted us to identify plant matrices containing AzA and to carry out a preliminary evaluation of the biological activities of extracts obtained through cheap and accessible extraction methods. In this respect, wheat (*Triticum durum*) was identified as an attractive matrix, due to its abundance, easy availability and safety profile. It is reported that cereals and grains, including barley, rye, wheat and sorghum, are sources of AzA [5]. In this paper, two samples of whole grain and whole-grain flour, respectively, of *Tricutum durum* from “Azienda Agricola Somma” (Catania, Italy), were selected, from which 17 different extracts were prepared. The aim of the study was therefore to obtain a moderate yield of AzA from vegetable sources and to evaluate the antioxidant activity using spectrophotometric assays (ABTS, DPPH, and Folin–Ciocalteu) and antimicrobial activity, by performing MIC assays against several pathogens. Our study aimed also to find a green alternative to the synthetically derived AzA used in cosmetics and pharmaceuticals.

## 2. Results and Discussion 

### 2.1. Analyte-Validation Method

The determination of analytes in plant samples depends greatly on the accuracy and sensitivity required and the interferences encountered in the sample matrix. As HPLC-MS is a suitable tool for the rapid and reliable determination of this type of compound, the chromatographic analysis was based on a previously reported HPLC-MS methodology [6]. The optimized analytical method (see experimental part) allowed the achievement of good separation of AzA, with a retention time at 11.40 (minutes), as can be seen in Figure 1. The LOD (limit-of-detection)/LOQ (limit-of-quantification) values were found to be 0.18 and 0.54 μg/mL, respectively (Table 1). The specificity of the proposed method was evaluated by analyzing processed aqueous solution, both blank and spiked with analyte AzA (Figure 2). Three concentrations were selected, and the residual standard deviation (%RSD) was calculated. In addition, the measurement was carried out three times on the same day to obtain intra-day variability and three times on different days to obtain inter-day variability. The results showed that the RSDs for AzA at concentrations of 300, 500 and 1000 ng/mL were 1.01%, 0.78%, 0.85%, respectively, for the intra-day measurement and 1.28%, 1.48%, 1.22%, respectively, for inter-day variability, as reported in Table 2.

### 2.2. Quantification of Azelaic Acid

The final conditions for sample processing were simple and fast. Solid samples were resuspended in methanol, and fluid samples were diluted (100-fold) with HPLC-grade methanol, given the sensitivity of the method and the moderate analyte concentration in the samples. It should be noted that sample dilution has some advantages, such as less matrix loaded into the column, thus reducing matrix effects of liquid samples, and compensating for possible variability between samples of different origins or variety. The quantification of Aza in samples was made by building a calibration curve of AzA standard solution between 100 and 1000 ng/mL and measuring the area of the relative peak. Detection was performed using multiple reaction monitoring (MRM), monitoring two transitions: the parent peak was *m/z* 187, which corresponds to the ion [M-H]^−^, the first transition was *m/z* 125 [M-OHOCO]^−^ with collision energy 17, and the second one *m/z* 97 [M-(OHOCO)_2_]^−^, with collision energy 20. The first transition reported was used as a quantification, while the second was used as a qualifier. Data were plotted in Excel, and a calibration curve was obtained as necessary for sample quantification (Table 1). Of the seventeen extracts that were analyzed, eleven of these presented a significant concentration of the AzA, while in the other extracts the quantification was below LOQ (Table 3). The overall results demonstrate that Naviglio ^®^ extract was the most concentrated, but the % *w/v* was still below the concentration of AzA used for cosmetic and pharmaceutical purposes.

### 2.3. Antioxidant Activity

Cereals contain unique phytochemicals found in the free, soluble conjugate, and insoluble bound forms; the various classes of phenolic compounds in cereals include derivatives of benzoic and cinnamic acids, anthocyanidins, quinones, flavonols, chalcones, flavones, flavanones, and amino phenolic compounds, and contain tocotrienols and tocopherol [7,8]. Our complex extracts are thus anticipated to have a potentially exploitable antioxidant profile which cannot be present in formulations of synthetically derived AzA. To demonstrate the antioxidant activity of our extracts, three different methods were used. Whole grains and whole-grain flours were screened for their DPPH radical scavenger capability, followed by the determination of total antioxidant capacity by the ABTS assay expressed as TEAC (Trolox equivalent antioxidant capacity), and by estimation of the total phenolic content, through the Folin–Ciocalteu assay. The measured DPPH and ABTS (expressed as TEAC values) activities for the extracts showed almost the same trend, with no clear antioxidant properties. The higher TEAC values were obtained with whole-grain and whole-grain-flour samples extracted by 70% hydroalcoholic maceration. The highest content of phenols was found in extracts with 70% hydroalcoholic ethanol solutions. In particular, the freeze-dried sample obtained by hydroalcoholic maceration showed TPC (total phenolic content) 91.40 μg/GAE. As for the whole-grain-flour extracts, the fluid-water extract presented the highest TPC with 69.48 μg/GAE, as shown in Table 4; in general, the TPC value was higher in flour extracts, probably due to the larger extraction surface.

### 2.4. Antimicrobial Activity

AzA is a well-known antimicrobial agent which exhibits relevant antibacterial and anti-inflammatory activities, and it is used as a routine treatment against acne vulgaris and rosacea. Furthermore, its use decreases the production of keratin, a natural substance that promotes the growth of acne bacteria, and it has shown a great potential for the treatment of other skin pathologies, such as malignant melanoma, hyperpigmentation and melasma. At the same time, our results demonstrated that pure AzA at a concentration of 256 µg/mL had no antibacterial activity against *S. aureus, E. coli and P. aeruginosa*, while it showed a MIC at 256 µg/mL against *Streptococcus pyogenes* (see Table 5). Results agree with data reported in the literature [9], where they found that AzA exerted its antibacterial properties at higher concentration, approximately at 16,000 µg/mL. Some information is available concerning the antifungal activity of AzA. Extracts obtained with hydroalcoholic maceration revealed optimal antibacterial activity, compared to the water extracts. Furthermore, data shown that the starting state of the vegetable matrix also affected the antimicrobial activity; in fact, whole-grain hydroalcoholic extract (entry #5,7) showed lower MIC values compared to the flour extract. As confirmation of better antimicrobial activity of whole grain, the lowest MIC value against P. aeruginosa was observed with the whole-grain maceration extract (entry #7). Evaluation of the activity of our extracts against fungal strains M. furfur and C. albicans yielded very good results, with MIC ranging from 5 to 0.176% (see Table 6) for the various samples. Interestingly these activities are not due to the presence of AzA. Indeed, in our hands, AzA was inactive up to 256 µg/mL against the two fungi, in contrast with previous reports [1,10]. In order to assess the impact of the preservative system (water and 0.05% sodium benzoate, 2% ethanol, citric acid to pH 4-4.5), MIC assays were performed on the three water-containing solvents (entry #3, 4, 6, Table 6). No intrinsic antimicrobial activity was observed, thus confirming that the observed antimicrobial activity of the extracts is due to the phytocomplex, not to the preservative system used. In the case of the Naviglio*^®^* extract (entry #8, Table 6), however, the MIC of the solvent system was comparable to that of the extracts, indicating that the antimicrobial activity of the Naviglio*^®^* extract is deeply affected by the solvent and preservative system (data not shown).

### 2.5. Multivariate Modelling and Data Analysis

Data analysis was carried out on the 17 × 10 matrix containing the 10 biological characterizations carried out on the 17 extracts (see Appendix A). PCA, a data display method used in multivariate analysis for the exploratory analysis of a multivariate dataset, was applied. The PCA performed on the data set identifies two significant components that explain 54.4% and 20.4%, respectively, of the total variance (74.8%), as reported in the scree plot (Figure 3).

As highlighted in Figure 4, PC1 mainly describes the extraction treatments; in particular, water extracts rich in AzA are separated from the others, with hydroalcoholic and water-extracted samples with low AzA content (in fluid state) with negative scores (left-hand side of the plot) and water samples rich in AzA with positive scores (right-hand side of the plot). On the other hand, it is clear that the samples are separated along the PC2 according to their bioactivity. Figure 5 displays the loading plot of the PCA. It is evident that variables such ABTS, DPPH and AzA concentration are highly correlated, and it is also easy to overview the relationship among the MIC values, except the MIC of *P. aeruginosa* which lies between antioxidant activity and the other MIC values. Analysis of the biplot (Figure 6) allowed us to identify variables which best discriminate among extraction techniques and sample matrices. For instance, AzA concentration, DPPH and ABTS clearly separate the fluid Naviglio sample and the fluid hydroalcoholic maceration from all the others. Seven samples are separated in the biplot. Sample 1 (the fluid sample obtained by water maceration from wheat grain), lies on the right side of the box plot and is characterized by good antimicrobial activity and moderate AzA concentration. Sample 7 (hydroalcoholic maceration) is the only one with a good MIC value against *P. aeruginosa* (as can be depicted by its position on the biplot). Sample 9 (fluid extract obtained using Naviglio technology, from wheat grain), presented the highest content of AzA, and it is located on the upper side of the plot. The sample with the best antimicrobial properties and AzA content was number 5 (fluid samples obtained from hydroalcoholic solution using ultrasound-assisted technology from wheat grain); in fact, it is located beyond the values. In addition, samples 14 and 16 (fluid samples obtained using hydroalcoholic maceration and ultrasound-assisted technology, respectively, from wheat flour), presented good antimicrobial properties, except against *P. aeruginosa* and moderate-to-low AzA content.

## 3. Materials and Methods

### 3.1. Chemicals and Reagents

HPLC-MS grade methanol and acetonitrile were purchased from Scharlab Italia Srl (Milan, Italy); bidistilled water was obtained using Milli-Q System (Millipore, Bedford, MA, USA). MS-grade formic acid from Fisher Chemical (Thermo Fisher Scientific Inc., San Jose, CA, USA) distilled water, ethanol and hydroalcoholic 70% solution were also used for the extraction. The 2,2-Diphenyl-1-picrylhydrazyl (DPPH), 6-hydroxy-2,5,7,8-tetramethylchroman-2-carboxylic acid (Trolox), 2,2′-azinobis(3-ethylbenzothiazoline-6-sulfonic acid) diammonium salt (ABTS), potassium persulfate, phosphate buffer, 2,4,6-tri(2-pyridyl)-s-triazine (TPTZ), iron (III) chloride hexahydrate, gallic acid, and Folin–Ciocalteu reagent were purchased from Sigma-Aldrich (Germany). Azelaic acid, an analytical standard, was purchased from Sigma-Aldrich.

### 3.2. Plant Material

*Triticum durum* samples, whole grain and stone-ground whole-wheat flour were purchased from “Azienda Agricola Somma” (Catania, Italy).

### 3.3. Sample Preparation

Whole grain and wheat flour (20 *g*) were processed in five different ways; for each extraction, the solvent-to-matrix ratio was 1:10. Each of the 17 extracts was filtered and divided into two different aliquots. One was kept in a fluid state and the other was lyophilized. The lyophilization process lasted 36 h, with −56 °C and 1.0 mbar using 1-DL Alpha Plus freeze-drier. The first extraction method consisted of traditional maceration, using water and a hydroalcoholic 70% solution, separately, as solvents. The dried matrix and solvent were placed in a backer and the extraction was stirred at 300 rpm for 2 h at room temperature. To avoid microbial contamination, the extracts were added with sodium benzoate 0.05%, ethanol 2%, and citric acid, until pH 4 was reached. The second extraction method adopted was ultrasound-assisted extraction using two different solvents: water and 70% hydroalcoholic solution. Similarly, the water and matrix were placed in a backer and sonicated at room temperature for 30 min; the water extract was stored as described above, while the hydroalcoholic extract, due to its ethanol content, is self-preserving. The alcoholic part of each extract was removed with a rotary evaporator. The third extraction method was the Naviglio^®^ extraction. For apparatus convenience, we decide to extract only the whole grain with this technology. Water, glycerin, and 1,3-butanediol (1:1:1 ratio) were used as solvents, with the same preservative system used for conventional extracts. Naviglio extraction lasted 40 extractive cycles, with 2 min in dynamic and 2 min in static phase. All the dried extracts were reconstructed with methanol HPLC-MS analysis.

### 3.4. Instrumentation

HPLC-MS analysis was performed using a 2695 Alliance separation system (Waters Go, Milford, MA, USA) equipped with a QuattroTM API triple quadrupole mass spectrometer with an electrospray source (Micromass, Water, Manchester, UK). Chromatographic conditions were the following: Column XSelect^®^HSST3 (250 mm × 2.1 mm, 5 µm), flow rate 0.2 mL/min, column temperature 30 °C, injection volume 10 μL. A gradient profile was applied using water (eluent A) and acetonitrile (eluent B) as mobile phases. Initial conditions were set at 100% A, after 5 min of the isocratic step, a linear change to 100% B at 8 min, and holding for 5 min before returning to initial conditions. Columns recondition was achieved over 6 min, providing a total run time of 20 min. The column was maintained at 30 °C and a flow rate of 0.20 mL/min was used. MSD parameter: ESI negative, capillary voltage 2.5 kV, cone voltage 25 V, extractor voltage 2, source block temperature 120 °C, desolvation temperature 350 °C. Cone-gas-flow nitrogen and argon were used as the collision gas. Detection was performed using multiple reaction monitoring (MRM) by monitoring two transitions; the parent peak was 187, which corresponds to the ion [M-H]^−^, the first transition was *m/z* 125 [M-OHOCO]^−^, with collision energy 17, and the second one *m/z* 97 [M-(OHOCO)_2_]^−^, with collision energy 20. The first transition reported was used as a quantification, while the second was used as a qualifier.

### 3.5. Preparation of Stock Solutions

The working standard AzA solutions were prepared by diluting the stock solution of AzA (1000 μg/mL) to a proper volume. The stock solutions were diluted to obtain working standard solutions in the concentration range from 100 to 1000 ng/mL.

### 3.6. Method Validation

The daily calibration standard was prepared by spiking 200 µL of blank aqueous solution with appropriate volumes of a standard solution of AzA to obtain a concentration of 100, 300, 500, 700, and 1000 ng/mL. The reliability of the HPLC-MS method was validated by its linearity, specificity, accuracy, precision, and stability.

#### 3.6.1. Linearity

To determine the linearity of the method, spiked standard samples at 5 concentrations over the range of 100 to 1000 ng/mL were prepared. The analysis was performed in three separate analytical runs for three sets of the abovementioned solutions. The calibration curve was constructed by adopting linear regression. The sensitivity of this HPLC-ESI-MS method was examined by the measurement of the LOD and LOQ. The LOD and the LOQ of the method were determined by the S/N, using the equation 3 S/N and 10 S/N, respectively [11]. The concentration range was obtained concerning the regression curve (y = ax + b) and correlation coefficient (R^2^).

#### 3.6.2. Specificity

The specificity of the proposed method was evaluated by analyzing an aqueous solution processed in both blank and spiked with the analyte.

#### 3.6.3. Precision

Inter- and intraday precision values were estimated by assaying a control aqueous solution containing three different concentrations of 300, 500, and 1000 µg/mL of AzA. The RSD was obtained by running the experiment three times in one day and repeating it on three separate days.

### 3.7. Evaluation of Antimicrobial/Antifungal Activity

The antimicrobial-activity evaluation of whole grain and flour was performed through a broth microdilution assay against *Escherichia coli* ATCC 25922, *Staphylococcus aureus* ATCC 25923, *Malassezia furfur* ATCC 14521, and *Candida albicans* ATCC 11006, *Pseudomonas aeruginosa* ATCC 27853 and *Streptococcus pyogenes* ATCC 19615. Minimal-inhibitory-concentration (MIC) evaluation was performed following the CLSI guidelines with some modifications (CLSI 2018b).

#### 3.7.1. Inoculum Preparation

Reference bacterial strains were inoculated in sterile Müeller Hinton Broth (MHB) and incubated overnight at 37 °C. Yeast strains were instead inoculated in sterile RPMI broth and incubated at 37 °C for 24/48 h. The bacterial/yeast suspension was then centrifuged for 20 min at 2000 rpm and 4 °C, and then the pellet was resuspended in phosphate buffer (PB). The bacterial suspension was adjusted in PB to obtain an optical-density (OD) value at 600 nm in a 1 cm light-path cuvette in the range 0.08–0.13, approximately equivalent to a 10^8^ CFU/mL suspension. The fungal suspension was adjusted to a final optical density of 0.5 McFarland standard (1–5 × 10^6^ cells/mL). The obtained bacterial and fungal suspensions were further diluted 1:100 in the appropriate medium to obtain a final concentration of 10^6^ CFU/mL, and inoculated within 30 min.

#### 3.7.2. MIC Assay

MIC assays were performed in 96-well microtiter plates incubating whole-grain and flour extract at concentrations ranging from 5 to 0.010% *v/v* with a final concentration of 5 × 10^5^ CFU/mL of bacterial suspension, in a total volume of 100 μL. MIC assays were conducted only for fluid extracts because the lyophilized ones showed a low content of AzA and low antioxidant capacity. MIC experiments were conducted also for pure solvents with a preservative system. The fluid extracts were tested in the range of 5–0.010% after filtration with a 0.22 nm filter to remove bacterial contamination. Ethanol was evaporated using a rotary evaporator. Pure AzA (in DMSO 2.56% *w/v*) was also tested at concentrations of 8, 16, 32, 64, 128 and 256 µg/mL, in order to discriminate the antimicrobial activity of AzA alone and the antimicrobial activity of AzA coupled with the phytocomplex. Growth and sterility controls were performed. For each test, three independent experiments, with three replicates each, were performed. After 24 h of incubation, the MIC value was evaluated as the arithmetic average of the lowest concentration of each compound that completely inhibited bacterial and/or fungal growth as detected by the unaided eye. The standard deviation from the average MIC value was also calculated. To evaluate the inhibition percentages of each tested concentration, the optical density (OD) of each plate was read with a spectrophotometer at 620 nm (data not shown, but available upon request to the authors). A quality-control organism (*E. coli* ATCC 25922) was tested periodically to validate the accuracy of the procedure.

### 3.8. Determination of Antioxidant Activity

The antioxidant activity of the different extracts against 2,2′-azino-bis(3-ethylbenzothiazoline-6-sulphonic acid) (ABTS) and 2,2-diphenyl-1-picrylhydrazyl (DPPH) was determined using a previously described method [12]. ABTS and DPPH are based on reactions with electron-donating or hydrogen radicals producing compounds/antioxidants. Electron transfer and hydrogen-atom-transfer reactions can be difficult to distinguish. Hydrogen-atom-transfer reactions can be the result of proton-coupled electron transfer. Despite the similar redox mechanisms of the methods, the reagents and products are different. In addition, performance and interpretation differ considerably among publications. Thus, reciprocal comparison of results of individual methods and results among publications for the same method is often problematic [13]. Trolox was used for the calibration of the TEAC and DPPH methods. For ABTS analysis, 30 µL of the sample (at different concentrations) was mixed with 2970 µL of ABTS reagent, already prepared from 5 mL ABTS solution 7 Mm and 88 µL of potassium persulfate 140 Mm, and analyzed at 734 nm against the blank. For DPPH analysis we used 100 µL of the sample (at different concentrations) and 2900 µL of DPPH reagent 0.05 Mm, prepared from dilution of the 1 Mm DPPH reagent solution. The growing concentration of extracts was prepared in order to build a concentration–response curve. The slope of the sample lines was used to calculate the TEAC (Trolox equivalent antioxidant capacity). Results were expressed as TEAC, which was calculated from the relation between the slope of the calibration curve and the slope of each plant extract. To obtain an estimation of the total phenolic content in each extract, the Folin–Ciocalteu colorimetric method was performed, and results were expressed as gallic acid equivalents (GAE), as previously reported. The Folin–Ciocalteu method, which is based on the reduction of a phosphonoformate−phosphomolybdate complex by phenolics to blue reaction products, was used to determine phenolic compounds [14]. For each antioxidant measure, the blank was made using pure solvent, e.g., for hydroalcoholic extracts the blank was the hydroalcoholic solution, for water extracts the blank was pure water with the preservative system.

### 3.9. Spectrophotometric Measurement

Absorbance measurements were taken using a Cary 60 UV-Vis (Agilent, Santa Clara, CA, 95051, United States) with a wavelength range of 190–1100 nm.

### 3.10. Multivariate Modelling

The most basic and often-used unsupervised chemometric approach is principal component analysis (PCA). Unsupervised principal components analysis (PCA) with autoscaling (mean centering and scaling) was performed to check the quality of the dataset. The starting point of PCA was the matrix of data with N rows (extracts) and K columns (variables; MS and biological data). The data set consists of 187 data: 17 rows and 10 columns. PCA helped to represent the multivariate data table as a low-dimensional space, consisting of two dimensions (as seen in the scree plot). This overview reveals groups of observations, trends and outliers and also uncovers the relationship between observation and variables and among the variables themselves. The number of PCs used, and hence the amount of variance collected, should be carefully chosen. PCA models were depicted as score plots, and consisted of two synthetic variables: principal component (PC) 1 (accounting for the greatest proportion of the total variance) and PC2 (accounting for the second-greatest proportion of the total variance orthogonal to PC1). These plots display intrinsic groups of samples based on variations. The score plot shows how the observation (extracts) are projected into the two-dimension space. Secondly, the other plot built was the loading plot, where loadings are the weights obtained from combining the original variables, to form the scores. Geometrically, they represent the direction of correlation in the space. The scores and the loadings are complementary and superimposable, which means that an interesting pattern seen in the scores plot can be interpreted by looking along the interesting direction in the loading plot [15]. Afterward, a biplot was built in order to understand the correlation between scores and loadings. Data analysis was performed using the Chemometric Agile Tool (CAT) (R. Leardi, C. Melzi, G. Polotti, CAT (Chemometric Agile Tool).

## 4. Conclusions

The obtained data show that water and hydroalcoholic wheat extracts obtained with simple maceration, ultrasound assistance, or Naviglio^®^ technology, contain variable amounts of AzA, a C9 dicarboxylic acid widely used for its antimicrobial and anti-inflammatory properties in cosmetics and dermatology. Even in the best obtained condition (Naviglio^®^ extraction, dried whole grains), the amount of recovered AzA is significantly lower (3% *w/v*) than that required for cosmetics (10% *w/v*) or pharmaceutical (20% *w/v*) use. Nevertheless, most of the obtained extracts showed very interesting properties in terms of antibacterial activities against common skin pathogens, and appreciable antioxidant properties, mainly due to a high TPC of the extracted phytocomplex. Considering the easy accessibility, low cost and wide traditional use, wheat-grain extracts surely deserve further investigation as a potential topical remedy for mild skin conditions, such as acne, rosacea, and hyperpigmentation.

## Figures and Tables

**Figure 1 molecules-28-02134-f001:**
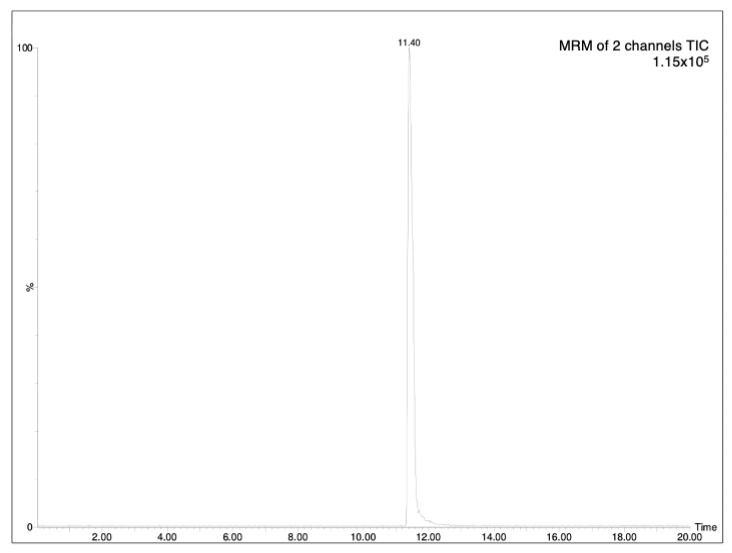
MRM of standard solution of AzA (500 ng/mL), Time is expressed in minutes.

**Figure 2 molecules-28-02134-f002:**
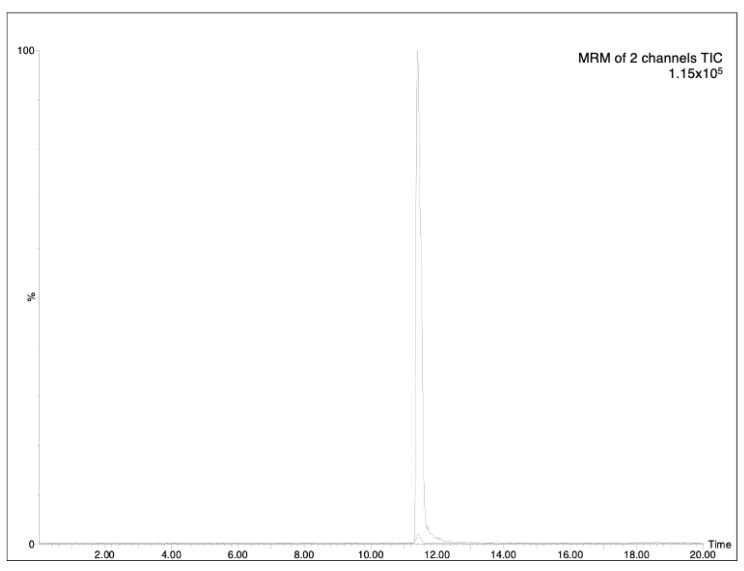
Specificity of the selected method, MRM of standard solution of AzA (500 ng/mL) against blank.

**Figure 3 molecules-28-02134-f003:**
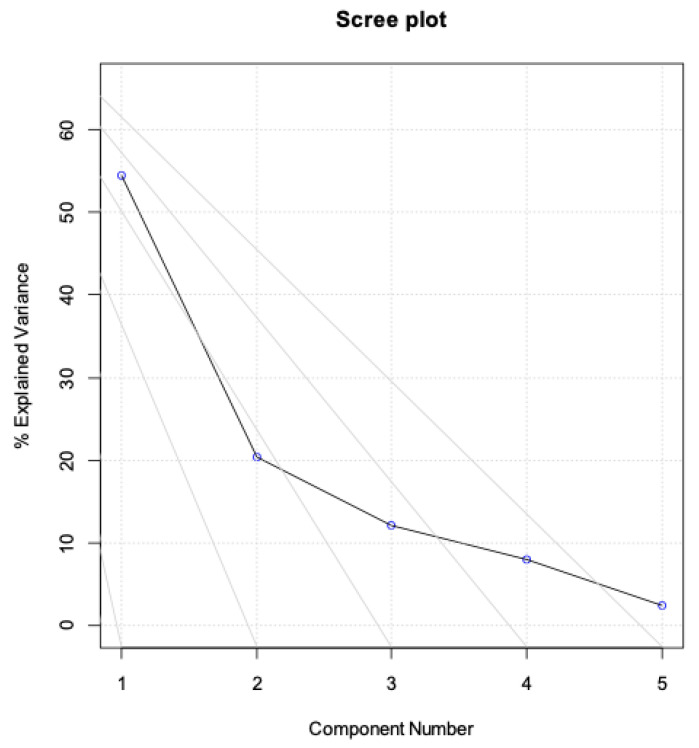
Scree plot of the data set (variance % explained by each component: 54.4; 20.4; 12.2; 8.0; 2.52).

**Figure 4 molecules-28-02134-f004:**
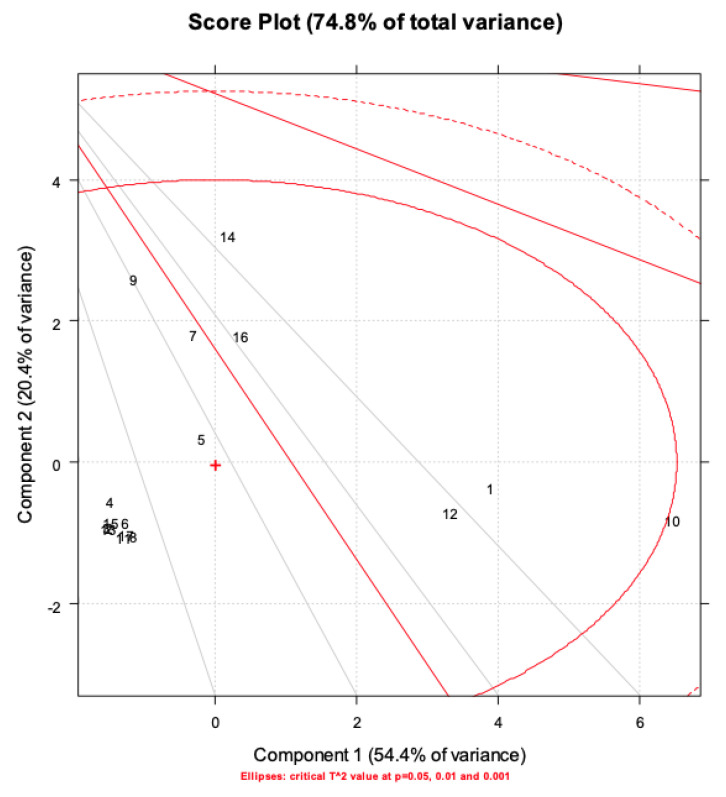
Score plot of PC1-PC2 (74.8% of total variance explained) of the data matrix. with the object coded according to the sample state (fluid or freeze-dried; respectively),and colored according to the starting material (whole grain or wheat flour, respectively).

**Figure 5 molecules-28-02134-f005:**
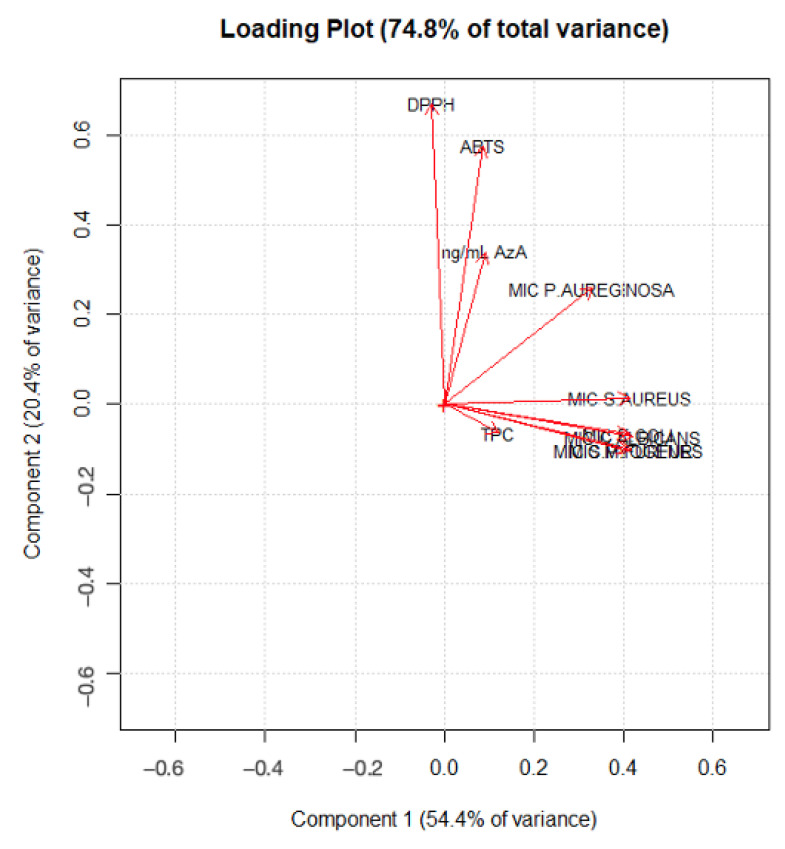
PCA loading plot of the first two principal components of the dataset.

**Figure 6 molecules-28-02134-f006:**
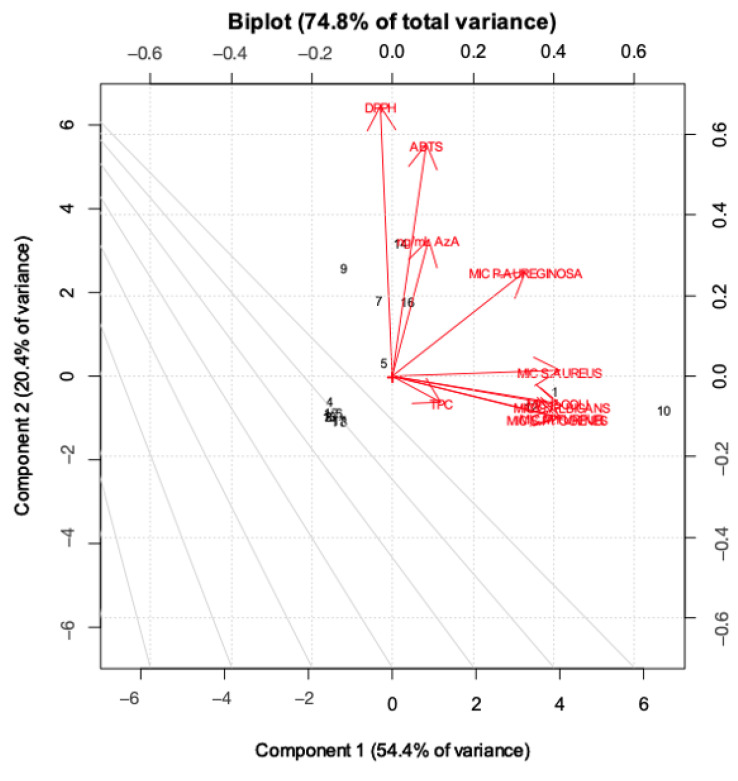
Biplot of the first two principal components, built using scores and loadings.

**Table 1 molecules-28-02134-t001:** Results of linearity regression, correlation coefficient, LOD, and LOQ for AzA.

	Regression Equation	R^2^	Calibration-Curve Slope	LOD (µg/mL)	LOQ (µg/mL)
**Azelaic acid** **Integration (n = 3)**	17.407x + 928.16 15.966x + 1452 17.454x + 954.97 16.942x + 1111.7	0.9977 0.9954 0.9929 0.9932	16.942	0.18	0.54

**Table 2 molecules-28-02134-t002:** Intra-day and inter-day variability of Aza.

		Intra-Day Variability (n = 3)		Inter-Day Variability (n = 3)	
**Standard**	Concentration (ng/mL)	Mean ± SD (Peaks Area)	*%RSD*	Mean ± SD (Peaks Area)	*%RSD*
	300	6182.33 ± 62.40	1.01	6157.66 ± 78.52	1.28
**Azelaic acid**	500	9525.66 ± 74.14	0.78	9490.66 ± 140.20	1.48
	1000	18801 ± 160	0.85	18.236 ± 222	1.22

**Table 3 molecules-28-02134-t003:** Results of Quantification of AzA in different wheat samples.

	Samples	Concentration (ng/mL ± SD)	*w/v* %
**Whole grain**	Water extraction (F)	9579.95 ± 74.47	0.96
	Water extraction (D)	<LOQ	/
	Water and ultra-sound extraction (F)	<LOQ	/
	Water and ultra-sound extraction (D)	<LOQ	/
	Hydroalcoholic 70% and ultra-sound extraction (F)	4711.96 ± 87.48	0.47
	Hydroalcoholic 70% and ultra-sound extraction (D)	2288.00 ± 75.95	0.23
	Hydroalcoholic 70% maceration (F)	5805.88 ± 68.45	0.6
	Hydroalcoholic 70% maceration (D)	1245.22 ± 100.24	0.12
	Naviglio (F)	34911.06 ± 79.66	3.4
**Wheat flour**	Water extraction (F)	9193.89 ± 98.12	0.92
	Water extraction (D)	<LOQ	/
	Water and ultra-sound extraction (F)	5927.87 ± 78.35	0.6
	Water and ultra-sound extraction (D)	<LOQ	/
	Hydroalcohlic 70% and ultra-sound extraction (F)	1510.84 ± 66.52	0.15
	Hydroalcoholic 70% and ultra-sound extraction (D)	1994.84 ± 78.03	0.20
	Hydroalcoholic 70% maceration (F)	1961.39 ± 100.55	0.20
	Hydroalcoholic 70% maceration (D)	<LOQ	

F: fluid extract; D: freeze-dried extract.

**Table 4 molecules-28-02134-t004:** Results of antioxidant activity in different wheat samples.

	SAMPLES	ABTS-TEAC	DPPH-TEAC	TPC (µg/GAE)
**Whole grain**	F Water extraction	0.402 ± 0.045	0.015 ± 0.004	6.02 ± 0
	D Water extraction	0.008 ± 0.000	0.002 ± 0	5.66 ± 0.01
	F Water and ultra-sound extraction	0.009 ± 0.001	0.001 ± 0	0.34 ± 0
	D Water and ultra-sound extraction	0.343 ± 0.054	0.026 ± 0.006	7.79 ± 0
	F Hydroalcoholic 70% and ultra-sound extraction	1.059 ± 0.021	0.017 ± 0.003	13.05 ± 0
	D Hydroalcoholic 70% and ultra-sound extraction	0.182 ± 0.010	0.004 ± 0	56.13 ± 0.01
	F Hydroalcoholic 70% maceration	3.643 ± 0.026	0.089 ± 0.005	22.1 ± 0
	D Hydroalcoholic 70% maceration	0.036 ± 0.013	0.003 ± 0	91.4 ± 0.01
	F Naviglio	0.678 ± 0.278	0.2 ± 0	10.48 ± 0.01
**Wheat flour**	F Water extraction	1.165 ± 0.041	ND	69.48 ± 0.03
	D Water extraction	0.001 ± 0.001	0.002 ±	60.39 ± 0.02
	F Water and ultra-sound extraction	0.506 ± 0.378	ND	66.03 ± 0.02
	D Water and ultra-sound extraction	0.009 ± 0.001	0.001 ± 0	7.99 ± 0.01
	F Hydroalcoholic 70% and ultra-sound extraction	3.572 ± 0.158	0.228 ± 0.004	52.27 ± 0.02
	D Hydroalcoholic 70% and ultra-sound extraction	0.044 ± 0.033	ND	7.99 ± 0.01
	F Hydroalcoholic 70% maceration	2.965 ± 0.052	0.103 ± 0.004	50.77 ± 0.04
	D Hydroalcoholic 70% maceration	0.068 ± 0.01	0.003 ± 0	66.84 ± 0.04

F: fluid extract; D: freeze-dried extract; ND: not detected.

**Table 5 molecules-28-02134-t005:** Results of MIC assay for azelaic acid expressed in µg/mL.

µg/mL	*E. coli* ATCC 25922	*S. aureus* ATCC 25923	*S. pyogenes* ATCC 19615	*P. aeruginosa* ATCC 27853	*C. albicans* ATCC 11006	*M. furfur* ATCC 14521
Azelaic acid	> 256 ± 0	> 256 ± 0	256 ± 0	> 256 ± 0	> 256 ± 0	> 256 ± 0

**Table 6 molecules-28-02134-t006:** Results of MIC assay expressed in *v/v* %.

Entry	Fluid Extracts (*v/v* %)	*E. coli ATCC 25922*	*S. aureus ATCC 25923*	*S. pyogenes ATCC 19615*	*P. aeruginosa ATCC 27853*	*C. albicans ATCC 11006*	*M. furfur ATCC 14521*
**1**	F HA maceration	0.781 ± 0.29	1.562 ± 0.58	0.156 ± 0	> 5 ± 0	0.469 ± 0.17	0.313 ± 0
**2**	F HA + ultra-sound	0.625 ± 0	0.937 ± 0.67	0.176 ± 0.05	> 5 ± 0	0.508 ± 0.33	0.313 ± 0
**3**	F W + ultra-sound	5 ± 0	2.5 ± 0	1.72 ± 0.65	5 ± 0	2.5 ± 0	2.81 ± 0.88
**4**	F W maceration	> 5 ± 0	5 ± 0	5 ± 0	> 5 ± 0	5 ± 0	5 ± 0
**5**	WG HA + ultra-sound	0.508±0,16	0.625 ± 0	0.156 ± 0	> 5 ± 0	0.391±0.14	0.176 ± 0.05
**6**	WG W maceration	> 5 ± 0	> 5 ± 0	2.97 ± 1.32	> 5 ± 0	> 5 ± 0	> 5 ± 0
**7**	WGHA maceration	0.703 ± 0.22	1.093 ± 0.65	0.313 ± 0	1.41 ± 0.44	0.313 ± 0	0.195 ± 0.07

F: extracts obtained from flour, WG: extracts obtained from whole grains, HA hydroalcoholic solution, W water.

## Data Availability

Data is contained within the article and Appendix A; data not shown can be requested from the authors.

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
