# Peer review of "Extraction and Quantification of Azelaic Acid from Different Wheat Samples (Triticum durum Desf.) and Evaluation of Their Antimicrobial and Antioxidant Activities"

_molecules, 2023, doi:10.3390/molecules28052134_

Round 1

Reviewer 1 Report

The paper deals with the extraction of azelaic acid (Aza) from different wheat samples (Triticum durum Desf.). A total of 17 different extracts were prepared, chemically analyzed and different concentrations of azelaic acid were detected. Also, the principal component statistical analysis was applied. In addition, the antioxidant and antimicrobial activity of the extracts were evaluated. The extracts exhibited low antioxidant and moderate antimicrobial activity.

The paper has significant methodology drawbacks (mandatory controls are missing). So, the paper needs major revision.

 Methodology drawbacks

Section 3.7.2 MIC assay

Please, give experimental confirmation that sodium benzoate 0,05%, ethanol 2%, and citric acid have not exhibited inhibitory effects on the tested bacterial and yeast strains. The sodium benzoate 0,05%, ethanol 2%, and citric acid were added to avoid microbial contamination (lines 252-253).

 Also, explain the effect of glycerin, and 1,3-butanediol (1:1:1 ratio, the Naviglio® extraction) used as solvents on the growth of the tested microorganisms.

 Line 334. Specify the working DMSO concentration.

 Lines 341-343 The inhibition percentages of each tested concentration are not presented in the Results and discussion.

 Section 3.8. Determination of antioxidant activity

 Line 361 Define the blank.

 Explain how working samples of fluid and lyophilized extracts at different concentrations for ABTS and DPPH assays were prepared.

 Please, explain how the preservative system used influenced the antioxidant activity of tested extracts.

 List of additional corrections

 The title should be changed since the antioxidant activity was also tested.

Lines 66-67 Please, cite the appropriate reference/s.

Lines 67-72 Rewrite and specify the aims of the study.

Lines 76-81 In my opinion, section 2.1 is not necessary for Results and discussion, it is explained in Materials and methods.

Line 90 Define the abbreviation - LOD/LOQ values (as a first appearance in the text).

Table 2. Add units for mean ± SD.

Line 153 Define the abbreviation - TPC values (as a first appearance in the text).

Lines 159 -161 The discussion of results is not in accordance with the results presented in Tab. 4. On the contrary, the TPC values of flour extracts were higher than those of whole grain extracts.

Line 162, Table 4. It should be Results of antioxidant activity of AzA in different wheat samples instead of Results of Quantification of AzA in different wheat samples.

Table 4. Add units for ABTS, DPPH, TPC.

Lines 171-180, Tab.5 Latin names should be italic (check throughout the text).

Line 178 Instead of 5 to 0.15% it should be 5 to 0.176% (Tab. 5).

Lines 176-181 Why did not explain the results of the antibacterial activity of the extracts against tested bacterial strains? The authors mentioned only antifungal activity.

Table 5. Add units for MIC values.

Table 5. In English, to write decimal numbers use a full stop, not a comma.

Table 5. For a better presentation, please, divide the extracts into two groups: flour extracts and whole grains extracts (the first column).

Reviewer 2 Report

The extraction of azelaic acid from different wheat matrices by different methods is reported. Azelaic acid was quantified by HPLC-MS. In addition to quantification, the researchers evaluated the antioxidant and antimicrobial potential, obtaining promising results. The concentration of azelaic acid is below the concentration required for use in pharmacology and/or cosmetics. 

Title:

OK (Adequate word count, clear and concise)

Abstract:  

A good contextualization is presented, the objective of the manuscript is clear, the methodology is addressed and some results are presented.  

Keywords:

Ok. They can increase to 10 words, remember that the greater the number of words, the greater the possibility of viewing the document.

Introduction

The authors make a good contextualization of the subject, knowledge gap is clear. The whole section is well referenced guaranteeing the scientific quality of the text. 

1) LineS 62, 64: write in the third person

Results and discussion

1) line 88, write in third person

2) It is understood that the unit of time is minutes, however, it is not superfluous to indicate the time units in line 89 and Figure 1. 

3) How did you ensure that the calibration curve concentrations were always the same i.e. in all three cases they were 100, 200, 300,...900 and 1000 ug/mL. Were they all developed from the same stock solution? 

4) Indicate the meaning of the acronyms LOD and LOQ (limit of detection  and limit of quantitation).

Materials and Methods

The methodology used is clearly described, which would allow replication of the research in other laboratories. The quality of the quantification method and the quality of the reagents used guarantee the quality of the results. 

Conclusions:  

1) line 397: writing in the third person

Round 2

Reviewer 1 Report

The authors made the necessary corrections, but there is an additional revision:

1. In the Introduction section, lines 64-65, for the statement "It is reported that cereals and grains, including barley, rye, wheat and sorghum, are sources of AzA." Please cite the appropriated reference/s

2. There is no need for ref. 5 "Azienda Agricola Somma” (Catania, Italy)". Delete from Reference List

3. Table 6. Whereas the authors confirmed that the solvent and preservative system from Naviglio® extract had an influence on the antimicrobial activity maybe the values for entry 8 should be excluded and in the text, line 198, add data not shown.

"In the case of the Naviglio® extract (entry #8, Table 6), on the contrary, the MIC of the solvent system is comparable to that of the extract, indicating that the antimicrobial activity of the Naviglio® extract is deeply affected by solvent and preservative system (data not shown)."

Also, the data for the Naviglio® extract should be excluded from PCA analysis.

4. Lines 221-223 Rewrite the sentence.

5. Lines 225, 231-232 If Naviglio® extract had no antimicrobial activity why discuss the results for this extract?

6. Line 236.  includeh includeincludee ?

7. Line 236 Define AOCs

8. Section 3.7.2 Line 357 It is preferable to use a % conc. of DMSO in antimicrobial testing.

9. Section 3.7.2 Please include that the preservative system and solvents were tested.

10. Check once again if the Reference List is prepared according to Journal's Instruction
